# Structural Malformations in the Neonatal Rat Brain Accompany Developmental Exposure to Ammonium Perchlorate [note 1]

**DOI:** 10.3390/toxics11121027

**Published:** 2023-12-18

**Authors:** Mary E. Gilbert, Katherine L. O’Shaughnessy, Kiersten S. Bell, Jermaine L. Ford

**Affiliations:** 1Centre for Public Health and Environmental Assessment, Office of Research and Development, Environmetal Protection Agency, Research Triangle Park, NC 27709, USA; oshaughnessy.katie@epa.gov; 2College of Pharmacy, The University of Texas at Austin, Austin, TX 78712, USA; kierstenbell@utexas.edu; 3National Center for Computational Toxicology, Office of Research and Development, Environmental Protection Agency, Research Triangle Park, NC 27709, USA; ford.jermaine@epa.gov

**Keywords:** perchlorate, brain, development, neurotoxicity, thyroid hormone, AOP

## Abstract

Environmental contaminants are often flagged as thyroid system disruptors due to their actions to reduce serum thyroxine (T4) in rodent models. The presence of a periventricular heterotopia (PVH), a brain malformation resulting from T4 insufficiency, has been described in response to T4 decrements induced by pharmaceuticals that reduce the hormone synthesis enzyme thyroperoxidase. In this report, we extend these observations to the environmental contaminant perchlorate, an agent that interferes with thyroid status by inhibiting iodine uptake into the thyroid gland. Pregnant rat dams were administered perchlorate in their drinking water (0, 30, 100, 300, 1000 ppm) from gestational day (GD) 6 until the weaning of pups on postnatal day (PN) 21. Serum T4 was reduced in dams and fetuses in late gestation and remained lower in lactating dams. Pup serum and brain T4, however, were not reduced beyond PN0, and small PVHs were evident in the brains of offspring when assessed on PN14. To emulate the developmental time window of the brain in humans, a second study was conducted in which pups from perchlorate-exposed dams were administered perchlorate orally from PN0 to PN6. This treatment reduced serum and brain T4 in the pup and resulted in large PVH. A third study extended the period of serum and brain TH suppression in pups by coupling maternal perchlorate exposure with maternal dietary iodine deficiency (ID). No PVHs were evident in the pups from ID dams, small PVHs were observed in the offspring of dams exposed to 300 ppm of perchlorate, and very large PVHs were present in the brains of pups born to dams receiving ID and perchlorate. These findings underscore the importance of the inclusion of serum hormone profiles in pregnant dams and fetuses in in vivo screens for thyroid-system-disrupting chemicals and indicate that chemical-induced decreases in fetal rat serum that resolve in the immediate postnatal period may still harbor considerable concern for neurodevelopment in humans.

## 1. Introduction

Thyroid hormones are critical for normal brain development, with severe hormone deficiencies during pregnancy or early life leading to a barrage of structural and functional deficits in the mammalian brain. A variety of environmental contaminants reduce circulating levels of thyroid hormones, most notably thyroxine (T4), raising concern for the potential impact of their exposure on children’s neurodevelopment. Our laboratory has worked to identify quantitative brain-based outcomes that may accompany modest insults to the thyroid system to better model environmental exposures. In the rat, we previously reported the presence of a periventricular heterotopia (PVH), a cluster of ectopic neurons in the posterior forebrain of animals born to hypothyroid dams. This malformation is reliably induced by exposure to the pharmaceuticals propylthiouracil (PTU) and methimazole (MMI) in our laboratory and others [1,2,3,4,5,6,7]. Additionally, its size is dose-dependently aligned with declines in serum T4, and once formed, a PVH remains in the brain for the life of the animal [1,2,3]. It has also been established that a PVH is a consequence of a disruption to radial glial scaffolding in the posterior forebrain accounting for the positioning of the observed clustering of ectopic neurons. Radial glia are a known target of thyroid hormone action, further strengthening the interpretation that the PVH represents a permanent indicator of TH-dependent abnormal brain development [4,5,8,9,10]. The presence of a PVH has been proposed as a simple brain-based readout of thyroid hormone signaling with potential application in regulatory settings [6,11,12,13,14,15,16]. However, to date, reports on PVH stemming from chemically-induced TH disruption have been largely restricted to the pharmaceuticals MMI and PTU, highly specific blockers of the hormone synthesis enzyme thyroperoxidase (TPO) [1,2,3,6]. Recently, the toxicological relevance of the PVH has been bolstered by a report of a PVH accompanying perinatal exposure to the TPO-inhibiting pesticide amitrole [17].

In this study, we sought to determine if an environmental chemical acting through a distinct mode of action from TPO inhibition would similarly result in a PVH. Here, we examined the effects of perchlorate, a well-known drinking water contaminant that reduces circulating levels of thyroid hormone in humans and animal models by blocking the sodium–iodide symporter (NIS) [18,19,20]. Inhibition of the NIS restricts iodine supply to the thyroid gland and curtails hormone synthesis [18,21,22]. Recent work from our laboratory confirmed previous reports of dose-dependent reductions in serum thyroid hormones in the pregnant dam and late-term fetus [15,22]. In the present report, we examined perchlorate-exposed pups on postnatal day (PN)14 for the presence of a PVH under three distinct experimental manipulations. Our findings demonstrate that a PVH can be induced in perchlorate-exposed animals, indicating that chemical action beyond TPO inhibition is sufficient to result in a permanent neurodevelopmental insult. This demonstrates the generalizability of this structural defect accompanying developmental thyroid disruption and supports its potential utility as a biomarker of thyroid-induced neurotoxicity in a regulatory setting. Finally, these observations underscore the need to consider chemical and non-chemical environmental factors that may exacerbate the toxicity of chemical exposures to the developing brain.

## 2. Materials and Methods

### 2.1. Experiment 1—Developmental Perchlorate Dose Response Assessment

Animals: Pregnant Long–Evans rats were obtained from Charles River (Raleigh, NC) on gestational (GD) 2 and housed individually in standard plastic hanging cages in an approved animal facility. All experiments were conducted with prior approval from the United States Environmental Protection Agency’s Institutional Animal Care and Use Committee (IACUC) and were carried out in an Association for Assessment and Accreditation of Laboratory Animal Care (AAALAC)-approved facility. Animal rooms were maintained on a 12:12 light/dark schedule, and the animals were permitted free access to food and water. On arrival, GD2 pregnant dams were individually housed and placed on a controlled but sufficient supply of iodine (225 µg/kg potassium iodate, D100001, Research Diets, Newark, NJ), with a mean daily iodine intake of 5.5 µg across gestation and lactation [23]. The drinking water source was deionized water. On GD6, the animals were weighed and weight-ranked across the 5 dose groups. From GD6 to PN21, the dams (n = 10–12/dose group) were administered 0, 30, 100, 300, or 1000 ppm of ammonium perchlorate (ClO_4_, CAS No. 7790-98-9, purity 99.5%, Sigma, St. Louis, MO, USA) in their drinking water. Dam body weights were monitored frequently throughout pregnancy, and the litters were weighed by sex on several days until the pups were weaned on PN21. One male pup was euthanized by decapitation on PN6 for gene expression of the heterotopia-forming region. One male and one female pup were sacrificed on PN14 for immunohistochemistry and the detection of a PVH, as described below.

### 2.2. Experiment 2—Postnatal Dosing with Perchlorate

Animals: In Experiment 2, 8 pregnant LE dams were administered 1000 ppm through drinking water beginning on GD6 and continuing throughout lactation. At parturition, litters were culled to 9 pups, 5 males and 4 females to the degree possible, and split across three intra-litter direct-pup perchlorate-dosing conditions. One dose level was chosen to emulate the dam’s 1000 ppm dose (~114 mg/kg/day equivalent), a higher dose of perchlorate (~285 mg/kg/day) and a vehicle (deionized water) control. Three pups in each litter were assigned to one of these three dose groups. Perchlorate solutions were administered orally via micropipette in a 1.5 µL/gm volume by touching the snout with the micropipette tip to initiate the suckling reflex and slowly dispensing the perchlorate solution into the mouth. With the dam remaining on a drinking water solution of 1000 ppm, the direct dosing of the pups began in the afternoon on PN0 and continued twice daily at 9:00 am and 4:00 pm on each subsequent day up to and including PN6. On the mornings of PN2 and PN6, one pup from each dose group was euthanized by decapitation prior to dosing to collect blood and brain samples for a hormone analysis. On PN14, the remaining pups were euthanized, and their brains were collected for anatomical evaluation of a PVH as described below.

### 2.3. Experiment 3—Dietary Iodine Deficiency and Drinking Water Exposure to Perchlorate

Animals: In Experiment 3, nonpregnant LE rats were administered an iodine-replete (D10001 Research Diets containing 225 ng iodine/gm of chow, and the same as in experiments detailed above) and an iodine-deficient (ID) diet (25 ng iodine/gm of chow) for a minimum of 4 weeks prior to breeding. Sperm positivity based on vaginal lavage was deemed GD0, and half of each group of newly impregnated dams was administered 0 or 300 ppm of perchlorate in the drinking water beginning on GD6. This design yielded 4 treatment groups: Con-Con, control diet and drinking water; ID-Con, ID diet and control drinking water; Con-ClO4, control diet and perchlorate in the drinking water from GD6-PN21; and ID-ClO4, ID diet and perchlorate drinking water. The designation before the hyphen represents the iodine nutritional manipulation initiated prior to breeding, and the designation after the hyphen represents the presence or absence of perchlorate in the drinking water beginning on GD6, with 10–15 dams in each treatment condition. Offspring from each litter were euthanized by decapitation on PN0 (at cull), PN2, PN6 and PN14, and blood and brain samples were collected for serum and brain hormone analyses. On PN14, the brains from one male and one female from each litter were reserved for immunohistochemistry and examination for PVH as described below.

### 2.4. Perchlorate Analysis in Serum

Direct dosing experiments were conducted to maintain the exposure level attained in the late-term fetus into the immediate postnatal period. Perchlorate measurements in serum were carried out to determine how close these values came to those observed in fetal serum. Perchlorate was extracted from serum according to methods modified from Oldi et al. [24]. A 50 µL aliquot was transferred to a 1.5 mL Eppendorf centrifuge tube and diluted with deionized water. Internal Standard (C^l18^O_4_) (10 µL of 25 µg/mL) was added to 0.5 mL of the diluent in a Vivaspin 500 centrifugal concentrator (30,000 molecular cutoff) and vortexed to mix. The samples were filtered via centrifugation at 3000× *g* for 2 min. An aliquot of the filtrate (50 µL) was diluted with deionized water (200 µL) in an autosampler vial for analysis and analyzed via liquid chromatography–mass spectrometry (LC/MS/MS) using an AB Sciex (Framingham, MA, USA) Exion AC UHPLC-Qtrap 6500+ Linear Ion Trap LC/MS/MS system.

### 2.5. Serum and Brain Hormone Analysis

Serum thyroid hormones: Total T4 and tri-iodothyronine (T3) were analyzed in serum as previously described [16,25] using the same LC/MS/MS system used for perchlorate analysis. Solvent-based calibration standards were used for quantitation over a range of 10–10,000 pg/mL in a 100 µL standard curve volume. Two ion transitions were monitored for each target analyte, qualitatively identified based on retention time relative to the internal standard and calibration standard and the ratio of the peak areas of the monitored ion transitions. The lower limit of quantitation (LLOQ) for each analyte was set to the concentration of the lowest calibration standard that gave an acceptable ion ratio and an acceptable recovery (100 ± 30%) of the spike amount. Each sample batch consisted of a method blank, a laboratory control sample (blank spike) and a continuing calibration verification sample prepared in solvent. The lower limit of quantification (LLOQ) for both T4 and T3 was 0.1 ng/mL.

Brain thyroid hormones: Frozen brain samples were weighed and transferred to a 15 mL centrifuge tube containing methanol and spiked with a mixed stable isotope solution containing ^13^C_6_-T2, ^13^C_6_-T3 and ^13^C_6_-T4, as described in Ford et al. [26]. Briefly, thyroid hormones were extracted using methanol, chloroform and an aqueous solution of 0.05% CaCl_2_ followed by nitrogen evaporation. Solid-phase extraction using an Evolute 50 mg, 3 mL CX SPE cartridge (Biotage, Charlotte, NC, USA) was performed in 4% formic acid and acetonitrile on a vacuum manifold. Thyroid hormones were eluted using 5% NH_4_OH in methanol and evaporated to dryness, and the residue was dissolved in a solution of 5% acetonitrile: 95% water with 0.1% formic acid. SPE reconstituted extracts were analyzed for thyroid hormones via stable isotope dilution LC/MS/MS.

### 2.6. Gene Expression in PVH Region by Quantitative Real-Time PCR

In Experiments 1 and 2, a posterior region of the forebrain where the PVH forms was collected on PN6 and stored in RNALater. The small tissue mass collected comprised the dorsal half of a posterior section of brain that encompasses the PVH and included the ventricular zone, dorsal neocortex and dorsal hippocampus in an approximately 0.5–1 mm thick coronal section corresponding to Plates 36–39 of 2nd Edition Paxinos and Watson rat brain atlas [27]. The section was placed in cold phosphate-buffered saline, and under a dissection microscope, the ventral portion of the brain section below the dorsal hippocampus was removed (as previously described [4]). Total RNA was extracted via TRIzol^®^ (Invitrogen, Waltham, MA, USA) according to the manufacturer’s protocol. RNA pellets were resuspended in nuclease-free H_2_O, and RNA concentrations were measured on a Nanodrop 1000 (Nanodrop Technologies, Wilmington, DE, USA). RNA samples were treated with DNase I (Promega, M6101 (Madison, WI, USA)) and quantified using the Ribogreen Quantitation Kit (ThermoFisher, R11490 (Waltham, MA, USA)). DNase I-treated RNA was reverse-transcribed with the ABI cDNA Archive Kit (ThermoFisher, 4322171), and 25 ng of equivalent cDNA was amplified in a 12 mL volume using ABI TaqMan Gene Expression Assays and ABI Universal Master Mix (ThermoFisher, 4304437). Amplification was performed on an ABI model 7900HT sequence detection system, using standard Taqman cycling parameters. All samples were run in technical duplicates. Molecular probes for the heterotopia-forming regions included 7 targets (Table 1) selected from previous work demonstrating their role in heterotopia formation [4]. Beta-2-microglobulin (*B2m*) was used as the endogenous control as it did not differ between control and treated groups in a single-factor ANOVA. Data were analyzed by relative gene quantification using the 2_ddCt method [28].

### 2.7. Periventricular Heterotopia Assessment

For all studies, to the degree possible, one male and one female from each litter was euthanized by decapitation on PN14, the brain was removed from the skull, and whole brain was immersion-fixed in 4% paraformaldehyde for one week before its transfer to deOlmos cryogenic protective solution, as previously described [1,3]. Each brain was sectioned coronally on a vibratome at a thickness of 60 µM, and 30 consecutive sections spanning a region of ~Plates 29 to 40 in Paxinos and Watson rat brain atlas [27] were processed for neuronal nuclei (NeuN) immunohistochemistry according to previously published methods [1,2,3,29]. In brief, sections were incubated overnight at 4 °C in a NeuN primary antibody (Millipore, MAB377, 1:2500 dilution (Burlington, MA, USA)), followed by a biotinylated secondary antibody (1:400) in conjunction with avidin-biotin amplification (Vectastain Elite ABC, Vector, Burlingame, CA, USA). A signal was detected using diaminobenzidine tetrahydrochloride (DAB) as the chromogen. Sections were mounted on gelatin-coated glass slides, dried, counterstained with cresyl violet and cover-slipped. Sections were imaged using an Aperio AT2 slide scanner (Leica Biosystems, Buffalo Grove, IL, USA), and each tissue section scored for a PVH using Aperio ImageScope [v12.4.6.5003] software. If a heterotopia was detected, the area of this region was calculated by tracing the perimeter of the NeuN-positive cell cluster, and the heterotopia volume was estimated according to standard stereological principles. Mean volume estimates were compared across treatments. Incidence estimates were also calculated based on a nominal cutoff value of 0.006 mm^3^ determined from an evaluation of historical control data and a more liberal value of 0.003 mm^3^ used in previous reports of heterotopia volume from our laboratory [2,3,29,30].

### 2.8. Statistical Analysis

All data were expressed as means ± standard errors of the mean (SEM). The statistical significance of treatment effects on thyroid hormone concentrations was assessed with a one-way analysis of variance (ANOVA) using SAS Version 9.2 (Cary, NC, USA). When a significant treatment effect was detected (*p* < 0.05), pairwise differences among groups were tested post hoc using Dunnett’s adjustment for multiple comparisons. For measures taken across time, repeated measure analyses were performed. Incidences of heterotopia were evaluated using two nominal threshold cutoffs, 0.003 mm^3^ and 0.006 mm^3^, and a χ^2^ analysis was performed. The threshold for mean fold-change in gene expression was set at 1.25, and a 5% false discovery rate was applied. To control for experiment-wise error, alpha was reduced to 0.02 dividing 0.05 by the square root of the number of targets examined (0.05/√7).

## 3. Results and Discussion

### 3.1. Experiment 1: Dose–Response—Serum and Brain Thyroid Hormones and Heterotopia

The experimental design for Experiment 1 is summarized in Figure 1A. Pregnant rats were administered one of 5 concentrations of perchlorate in the drinking water from early gestation. Perchlorate reduced T4 in the sera of pregnant dams and pups but to varying degrees depending on the age and dose. Dose-dependent declines in serum T4 were evident in dams across pregnancy and lactation, with GD20 showing the most prominent reductions in T4 of >50% at the two highest dose levels (Figure 1B). In the newborn pups, significant declines in serum T4 were observed at the highest dose level, but these had recovered to or exceeded control levels by PN2, despite continued exposure to the dam (Figure 1C).

Mirroring very closely the results of serum, whole brain T4 and T3 in the late-term fetus (GD20, taken from [15]) and newborn pup (PN0 this study) were significantly and dose-dependently reduced, with recovery to control levels by PN2. Reductions in brain T3 and T4 were limited to the two highest doses of 300 and 1000 ppm of perchlorate (Figure 2). Although it did not reach statistical signficance, it is interesting to note that an increase in brain T4 was observed at the same dose for which signficant increases were also evident in serum T4 (Figure 1C and Figure 2A, 300 ppm on PN2).

Heterotopia were evident in the brains of male and female offspring when examined on PN14 (Figure 3) but, overall, they were relatively small compared to previous reports with PTU [1,2,7]. Quantification of mean PVH volumes in male and female offspring are presented in Figure 3B and 3C, respectively. The maximal mean volume of 0.008 mm^3^ was observed in males of the high-dose group, its small size contrasting to the order-of-magnitude-larger volumes typically observed in animals exposed to a moderate dose of the goitrogen PTU [2]. For reference, the inset in Figure 3C presents the same volume data on a y-axis coordinate typical of maternal exposure to a moderate 3ppm dose of PTU. Nonetheless, the results of the ANOVA summarized in the Figure 3 legend support the presence of a PVH at the 1000 ppm dose level of a magnitude exceeding noise levels detected in controls in both male and female offspring. The results of this analysis also indicated a marginally significant main effect of sex (*p* < 0.0456) suggestive of a slightly smaller heterotopia volume in the brains of female offspring relative to males, with no dose X sex interaction (*p* > 0.82). Despite the small PVH present in high-dose-perchlorate pups, using a liberal cutoff volume of 0.003 mm^3^ as a threshold criterion for heterotopia presence, incidence values of 50% and 72% relative to a control of 20% suggest the induction of a heterotopia in the 300 and 1000 ppm offspring (χ^2^ (4)= 23.2, *p* < 0.0001, N = 88). The application of a stricter heterotopia volume threshold of 0.006 mm^3^ reduced the incidence to 20% and 45% in the two highest dose groups relative to a 0% incidence in controls (χ^2^ (4) =17.87, *p* < 0.0013).

The relative expression levels of gene targets previously identified in the PVH-forming region and ventricular epithelium of the PTU-exposed PN6 neonatal brain [4,5] were not changed by perchlorate at any dose level (Figure 4). The lack of an effect of perchlorate on gene expression in the PVH-forming region, although failing to support the anatomical observations, are consistent with the minor size of PVH when present, the relative crudeness of the dissection of the region, and the limited postnatal serum and brain T4 reductions evident in offspring from perchlorate-exposed dams.

### 3.2. Experiment 2: Direct Postnatal Dosing—Thyroid Hormones and Heterotopia

Given observations from Experiment 1 in which serum and brain thyroid hormones recovered to control levels in the immediate postnatal period and PVH were small, we reasoned that extending the period of TH deficiency until at least PN6 may be necessary to more robustly induce a heterotopia. Lactational transfer can be limited for some thyroid-hormone-system-disrupting chemicals, and direct dosing has previously been implemented to extend the exposure to the pups into this period of brain development [31,32,33,34,35]. Previous work from our laboratory indicated that exposure to PTU that reduced serum T4 until at least PN6 was required for a PVH to form [1,3,4]. To accomplish a more prolonged postnatal TH insufficiency in the pups than was achievable with maternal perchlorate exposure alone, offspring born to dams exposed to 1000 ppm of perchlorate in drinking water beginning on GD6 were supplemented with perchlorate given orally over the first 6 days of life (Figure 5A). Perchlorate concentrations in the serum of pups from 1000 ppm dams supplemented orally with perchlorate exceeded those from maternal exposure alone, and by PN6, they had attained levels comparable to those observed in the fetus on GD20 in response to maternal exposure to 1000 ppm (Figure 5B, left vs. right). As such, the protocol was effective in maintaining “fetal levels of exposure” throughout the first week of life.

This direct dosing protocol also resulted in significant reductions in both serum and brain T3 and T4 relative to pups indirectly exposed from dams through lactation (Figure 5C). When assessed on PN14, this manipulation resulted in the production of a very large PVH in the offspring (Figure 5D,F). Small PVHs were also present in the offspring of 1000 ppm dosed dams who served as vehicle controls in this study, replicating observations in Experiment 1 (compare Figure 3B and Figure 5D). The PVH volume from non-exposed control pups (0 ppm dose group of Experiment 1) is included in the volumes summarized in Figure 5D to highlight the consistency of the presence of a small PVH in response to a high maternal dose of perchlorate. In orally supplemented pups, PVH volumes were increased 3–4-fold over the basal levels found with 1000 ppm maternal dosing. The exacerbation of heterotopia volumes with the continued oral dosing of pups is further supported by a significant downregulation of the expression of thyroid-hormone-responsive genes *Hr*, *Klf9* and *Shh* in the heterotopia-forming regions in pups on PN6 (Figure 5E, compare to Figure 4). A significant upregulation was seen in the transcript encoding brain-derived neurotrophic factor, *Bdnf_t_*.

Observations from the direct-dosing study support previous reports with PTU indicating that the reduction in serum and brain hormones must be extended to the early postnatal period in order for a PVH to form [1,3,4]. They further indicate that the presence of a large PVH is not limited to chemical actions at the TPO site, extending observations to a different mode of chemical action to inhibit NIS. The findings with perchlorate also support the role of thyroid hormones in neuronal migration and the impact of its deficiency on processes reliant on radial glia, as demonstrated in other developmental models of thyroid insult [3,5,8,9,10]. Together with the results of Experiment 1, the data support the use of this readout of thyroid-hormone-dependent neurotoxicity while underscoring the importance of considering chemical kinetics and life-stage sensitivity in interpreting data from animal studies. Internal dosimetry for perchlorate in these studies will serve to improve predictions in existing computational models of NIS inhibition and thyroid hormone disruption [36,37,38,39,40,41] and are the subject of ongoing investigations.

### 3.3. Experiment 3—Dietary Iodine Deficiency and Perchlorate Exposure

Distinct from Experiment 2, in this study, we set out to extend serum thyroid hormone insufficiencies induced by perchlorate exposure in a more physiological and controlled manner by coupling a moderate dose of perchlorate (300 ppm relative to 1000 ppm used in Experiment 2) with dietary iodine deficiency. The experimental design is summarized in Figure 6A. As depicted in Figure 6B, dam serum T4 during gestation and at the end of lactation was reduced by 300 ppm of perchlorate and by ID, findings consistent with previous reports from our laboratory [15,21,42,43] and the results of Experiment 1 (see Figure 1B). Greater deficits in serum T4 were evident in pregnant and lactating dams exposed jointly to prenatal ID and perchlorate beginning on GD6.

In contrast to dams, neither dietary ID nor perchlorate alone reduced thyroid hormones in the serum or brain in their offspring on PN0 or PN2. These results replicate the findings of Experiment 1 in which no reductions in pup serum T4 (Figure 1C), brain T4 (Figure 2A) or brain T3 (Figure 2B) were evident on PN2 at the 300 ppm dose level of perchlorate. However, they counter those of Experiment 1 for the day of birth (PN0). In the present study, there was no effect on pup serum or brain thyroid hormones on PN0 (Figure 6D,E), a dose that did reduce both serum and brain T4 in Experiment 1 (compare 300 ppm group in Figure 1C, Figure 2A at PN0 with Figure 6D,E). It is unclear what contributed to the differential effects on serum T4 at this intermediate perchlorate dose level at this young age in these two studies, both conducted in our laboratory and about 1 year apart. Three main differences between the two studies are breeding facility (Charles River Laboratories vs EPA), timing of animal shipment (GD2 vs. many weeks prior to breeding), and age at time of breeding (~PN50 to >~PN80-125); the latter varied as a necessity to establish steady-state iodine levels prior to breeding in Experiment 3 [23,42].

Despite these slight differences across studies, the present data reveal that the maternal administration of a seemingly non-effective dose of perchlorate alone for thyroid hormone effects in the pup, when administered to an iodine-deficient dam, greatly exacerbated the serum T4 reductions recorded for both dams and offspring on PN0 through PN14 (Figure 6B,C). Brain T3 and T4 were also reduced on PN0 in the pups from ID dams administered perchlorate, with no differences from the control seen when either treatment occurred alone. Interestingly, brain T3 reductions were restricted to PN0, with no difference from the control apparent in brain T3 at PN2, although a significant amount of variability surrounded values for brain T3 and T4 at this age (Figure 6D,E). These alterations in serum and brain thyroid hormones restricted to pups born to ID dams exposed to perchlorate were accompanied by large PVHs (Figure 7) exceeding volumes achieved in the direct dosing procedures of Experiment 2 or in response to a moderate dose of PTU (3 ppm) reported in previous studies [1,3,4,29,44].

No evidence of a differential effect of sex on PVH volume was seen, failing to support the marginal effect of greater sensitivity in males observed in Experiment 1. Collapsing across sex to calculate PVH incidence (N = 87) based on the liberal 0.003 mm^3^ cutoff resulted in comparable values in the Con-Con and ID-Con groups, 9.5% and 13.6%, respectively, with 42% of Con-ClO_4_ animals and 100% of ID-ClO_4_ animals exceeding this threshold (χ^2^ = (3) = 47.9, *p* < 0.0001), values remarkedly similar to those observed in Experiment 1. The insets in Figure 7 are also consistent with the trend observed in Experiment 1 for heterotopia volume, suggestive of increases above background for the 300 ppm dose level.

## 4. General Discussion and Conclusions

In this series of studies, we demonstrated the induction of PVH in animals exposed to perchlorate. This is the first report of this brain defect by an environmental contaminant that has a molecular initiating event outside of TPO inhibition. These findings are significant as they expand the concern of a thyroid-dependent neurodevelopmental insult from chemicals that target TPO to include environmental toxicants that also inhibit NIS.

In the dose–response analysis of perchlorate exposure, serum hormones were reduced in the perchlorate-exposed dams throughout gestation and lactation, but hormones in the nursing neonates were reduced only transiently or not at all. These findings support previous observations from our laboratory with this strain of rat [21] but for which iodine in the control diet was much greater than in the present study (i.e., >1000 ng/gm vs. 225 ng I/gm of chow in the current study) [21,43]. We had anticipated that a replete yet controlled and limited supply of iodine in the chow in the present study may have increased sensitivity to detecting an effect of ID on serum T4 in the neonate. This appeared not to be the case; no greater hormone sensitivity to perchlorate was observed in the present study relative to our previous report [21]. Nonetheless, the findings from both studies indicate that despite continued maternal exposure, significant reductions in the transfer and/or bioavailability of perchlorate occurs in the neonate relative to the fetus, as previously proposed by Clewell et al. [37]. This action contributes to the rapid recovery of thyroid hormones in the early postnatal period. Postnatal recovery to euthyroid status also occurs despite higher concentrations in milk samples from pups on PN2 than in the placenta in late gestation following a similar regimen of maternal perchlorate exposure [45]. The latter study also evidenced considerable resilience of the neonatal relative to the fetal thyroid gland. This took the form of a profound increase in relative *Nis* expression in the thyroid glands of pups on the day of birth (24-fold increase, see [45] relative to that seen in the late-term fetus (5-fold increase [15]). Together, these actions of curtailed exposure and the enhanced responsiveness of the neonatal thyroid gland by PN2 appear to effectively reverse the profound thyroid hormone suppression present in the late-term fetus. The significant increase in serum T4 observed on PN2, echoed in brain T4, may also reflect the interaction of augmented *Nis* expression in the gland in the face of falling perchlorate levels in the serum as the thyroid system of the very young neonate attempts to achieve homeostatic balance.

The direct dosing of pups with perchlorate or a combinatory maternal perchlorate exposure and marginal ID diet further establishes that the suppression of serum and brain thyroid hormones, if extended beyond PN0, has very marked consequences on brain development. These observations support and expand our previous work on the developmental timing of thyroid hormone insufficiency requirements for PVH formation and further implicate perchlorate as a developmental neurotoxicant [21]. Moreover, the augmented response in the brain to the pairing of two treatments, which independently had rather limited impacts on hormonal status or brain morphology, illustrates the importance of considering additional chemical and non-chemical environmental stressors in the evaluation of chemical hazard and estimation of risk.

Regulatory Implications: Differences exist in the timelines of rat and human maturation of the thyroid system and the brain with important implications for extrapolating from animals to humans, especially for effects on neurodevelopment. Although both mammalian species follow a similar ontogenetic trajectory, much of the brain development beginning at the end of the second trimester of human pregnancy occurs in the first few postnatal weeks in the rat [46,47,48]. In our studies, thyroid hormone insufficiency in the late gestation and immediate postnatal period is necessary for the PVH to form [1,3,4]. If thyroid hormone insufficiency in humans were to similarly lead to the migration errors that underlie the PVH, this period of sensitivity would be manifested in the fetus during the last trimester of pregnancy. This structural defect may not take an identical form or position in the human brain, but neuroimaging by Rovet [49] suggests the presence of an abnormality in the white matter in children born to hypothyroid women. Rodent models interrogating the potential impact of chemical exposures on brain occurring in the perinatal window in the rat suffer a major inconvenience of the timing of birth. Birth drastically alters the chemical exposure scenario from dam/placenta/fetus to lactational exposure to the pup, complicating extrapolation to humans and possibly reducing the sensitivity of the rodent model to detect brain insult. Here, we demonstrated the augmentation of a brain defect induced by an environmental contaminant, from being barely detectable to massively expressed, simply by ensuring that the time of hormone insufficiency encompassed both the late fetal and early neonatal period—a period of human thyroid-hormone-dependent brain development that occurs in the womb.

These observations have significant implications for the interpretation of regulatory studies of thyroid system disruption. Chemicals that reduce maternal thyroid hormones can limit hormone availability to the fetus. Chemicals that reduce maternal thyroid hormones but also gain access to the fetal compartment, threatening fetal hormone supplies, present an additional risk to the developing brain. Furthermore, test substances that are not readily transferred in the milk but that do perturb serum hormone profiles in the fetus may go undetected in a number of regulatory guideline studies in which hormonal screens are limited to postnatal ages beyond PN2 [50,51]. As such, our findings underscore the importance of the inclusion of serum hormone profiles in the pregnant dam and the fetus in in vivo screens for thyroid-system-disrupting chemicals and the merits of the EPA’s Comparative Thyroid Assay [52]. They further indicate that chemical-induced decreases in fetal rat serum that resolve in the immediate postnatal period, despite the absence of a robust marker of brain insult such as the PVH, may still harbor considerable concern for neurodevelopment in humans. The incorporation of hormones or markers of thyroid hormone action in the fetal brain at these critical ages into standard toxicity testing could provide critical information to reduce uncertainties in predicting the potential risk of developmental neurotoxicity from exposure to thyroid hormone-system-disrupting chemicals.

## Figures and Tables

**Figure 1 toxics-11-01027-f001:**
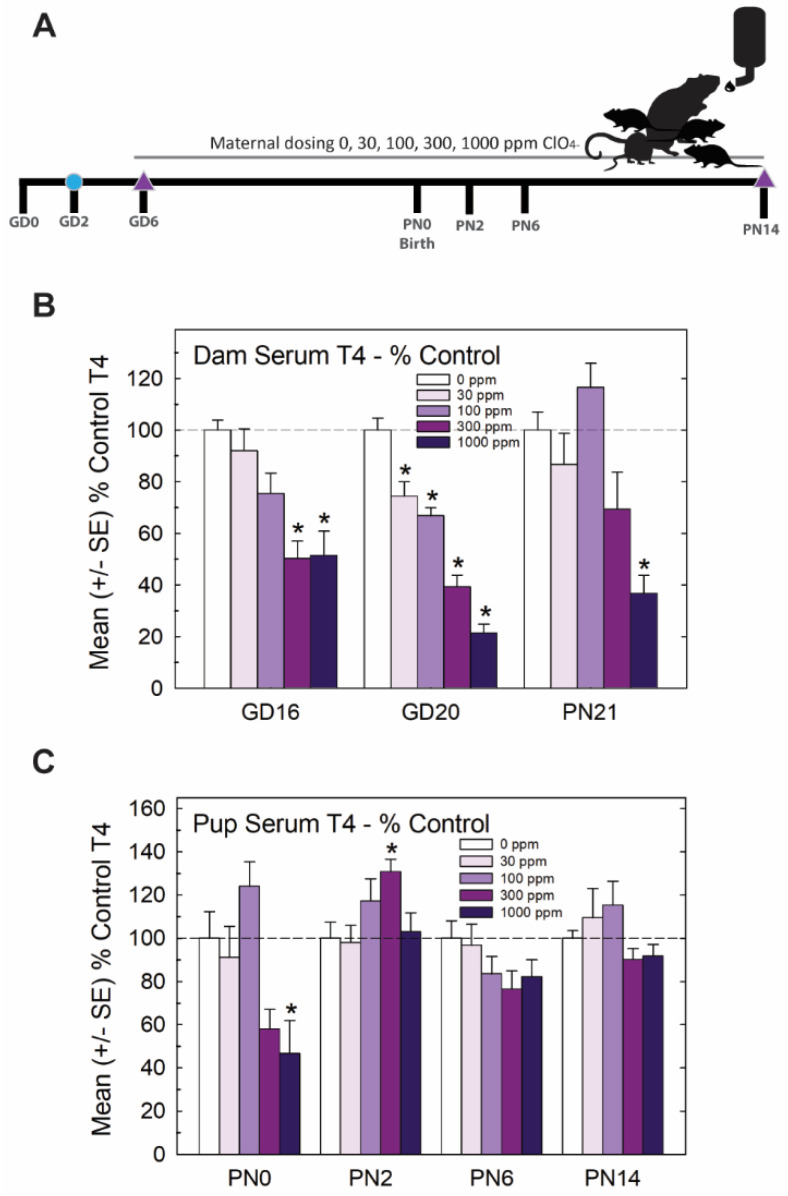
Experiment 1—developmental perchlorate dose–response assessment. (**A**) Experimental design. Animals were placed on an iodine-sufficient but controlled diet upon arrival to the facility on GD2. Maternal dosing with perchlorate via drinking water began on GD6 and continued until pups were weaned on PN21. Blood samples were collected from dams on GD16 and GD20 and at sacrifice on PN21 (not shown in schematic). Pups (1–2 pups/litter) were euthanized on the day of birth, PN2, PN6 and PN14 and blood was collected for serum TH analysis. The brain was collected from 1 male and 1 female pup from each litter on PN14 and processed for immunohistochemistry for PVH. (**B**) Dam serum T4 was dose-dependently reduced by perchlorate in mid- and late- gestation (GD16 and GD20) and on PN21 (n = 6–9/dose group). An overall ANOVA revealed significant main effects of dose (F(4,102), *p* < 0.0001), age (F(2,102) *p* < 0.0003), and dose X age interaction (F(8,102) = 2.65, *p* < 0.0112). Step-down ANOVAs at each age were significant (all *p*’s < 0.0003). On GD20, dam serum T4 concentrations were reduced at all dose levels but only the higher doses at earlier or later timepoints. * *p* < 0.05 by Dunnett’s *t*-test. (**C**) Pup serum T4 levels were only moderately impacted by maternal perchlorate exposure (n = 6–12/dose group). An overall ANOVA revealed significant main effects of dose (F(4,156) = 4.48, *p* < 0.0019) and age (F(3,156) = 9.19, *p* < 0.0001), and a dose X age interaction (F(12,156) = 3.31, *p* < 0.0003). Step-down ANOVAs at each age demonstrated a significant effect of dose at PN0 (F(4,37) = 5.87, *p* < 0.0009), with significant difference from the control by Dunnett’s post hoc *t*-test limited to the high-dose group (* *p* < 0.05). A significant effect of dose at PN2 (F(4,39) = 3.05, *p* < 0.02) also supported an effect of perchlorate on serum T4 but one that was primarily driven by an increase and not a decrease in T4 at the 300 ppm dose level. No main effect of dose was evident at the remaining two older ages (both *p*’s > 0.17).

**Figure 2 toxics-11-01027-f002:**
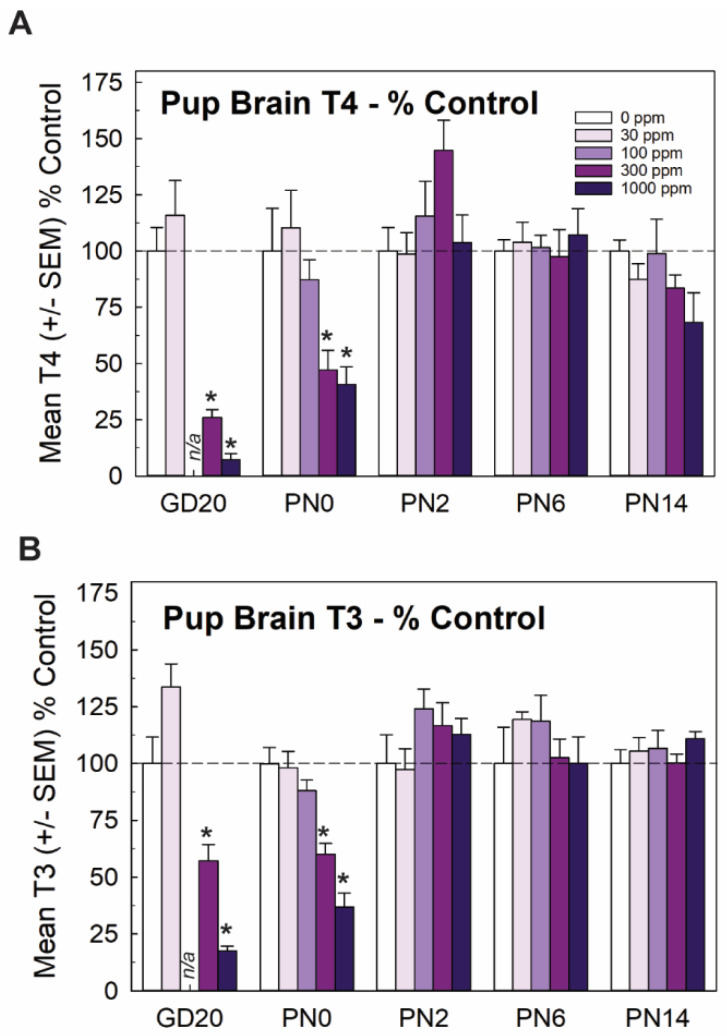
Experiment 1. Brain thyroid hormones in offspring. (**A**) Thyroid hormones were reduced in the brain of the newborn pup (PN0) at the two highest dose levels but did not differ from controls at ages of PN2 and beyond (n = 5–9/dose group). Fetal brain hormones were reported in Gilbert et al. [15] and are included here for completeness. An overall ANOVA for brain T4 at the postnatal timepoints revealed significant main effects of dose (F(4,101) = 2.71, *p* < 0.0343), age (F(3,101) = 9.99, *p* < 0.0001) and a dose X age interaction (F(12,19) = 3.21, *p* < 0.0006). Step-down ANOVAs revealed significant reductions at PN0 (F(4,32) = 5.74, *p* < 0.0013) and PN2 (F(4,31) = 2.71, *p* < 0.0484) that were restricted to the two highest dose levels. (**B**) A similar pattern is evident for brain T3, with an overall ANOVA supporting significant main effects of dose (F(4,101) = 4.10, *p* < 0.004), age (F(3,101) = 20.69) and a dose X age interaction (F(12,101) = 4.214, *p* < 0.0001). Significant reductions in brain T3 were seen in GD20 fetus [15] and pups on PN0 (F(4,33) = 21.06, *p* < 0.0001) with significant reductions at the two highest dose levels. No differences in brain T3 were observed at any other age. n/a not available; n = 5–9/dose group. * Dunnett’s t *p* < 0.05.

**Figure 3 toxics-11-01027-f003:**
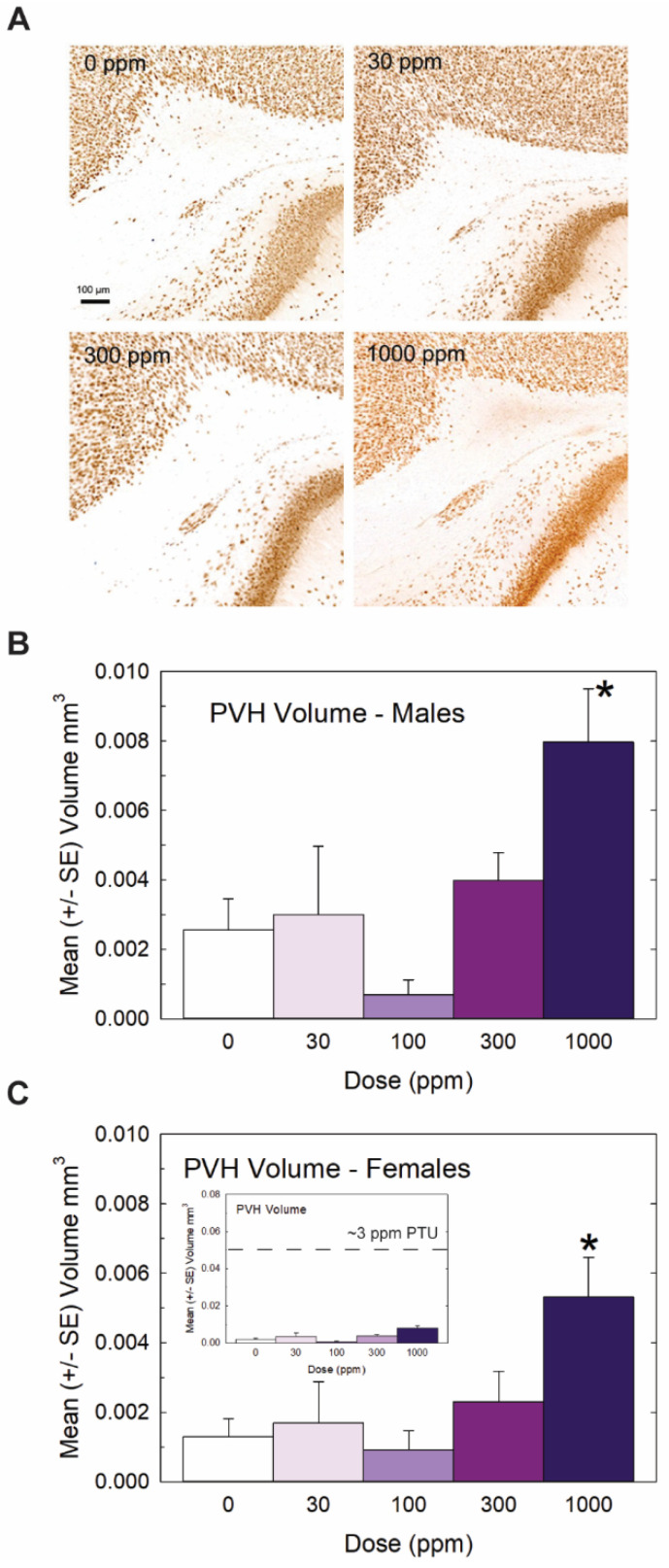
Experiment 1: Periventricular heterotopia (PVH) volume. Small PVH were present in the offspring of exposed dams and were restricted to the highest dose level. (**A**) Representative photomicrographs of a PVH in a coronal section of a PN14 rat brain. Volumes were calculated from area measures summed across all sections displaying a PVH in male (**B**) and female (**C**) offspring. An overall ANOVA revealed a significant main effect of dose (F(4,78) = 7.65, *p* < 0.0001) and a marginal effect of sex (F(1,78) = 4.13, *p* < 0.0456) but no dose X sex interaction (F(4,78) = 0.32, *p* > 0.83). Step-down ANOVAs by sex supported PVHs above background levels in males (F(4,35) = 4.12, *p* < 0.0077) and females (F(4.43) = 3.52, *p* < 0.0142) restricted to the 1000 dose level. Although PVHs were consistently observed at this high dose of perchlorate, the inset in (**C**) displays its relative minuteness when presented on a typical scale of PVH observed in previous studies with PTU (see [2,3]). N = 7–10 litters/dose group in males, N = 7–12 litters/dose group in females. (* Dunnett’s *t*-test, *p* < 0.05).

**Figure 4 toxics-11-01027-f004:**
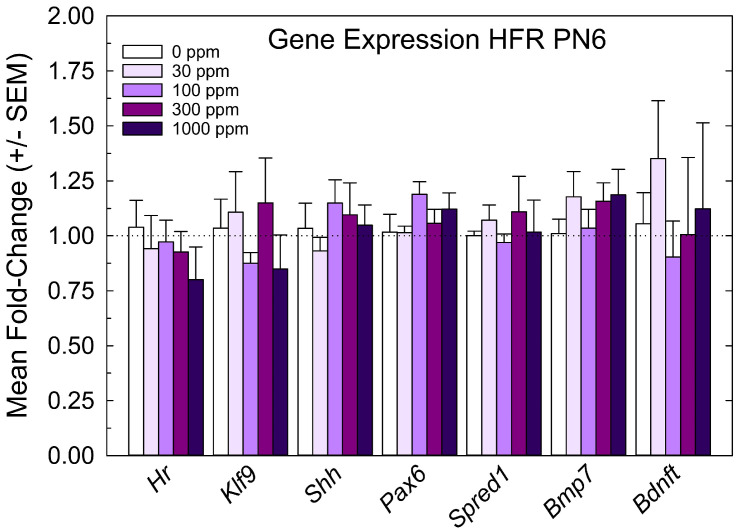
Experiment 1—Maternal exposure to perchlorate dose response assessment. Gene expression in the PVH-forming region of a PN6 brain. TH-responsive genes previously identified in a PVH-forming region were not changed in male offspring on PN6 (all *p*-values for all genes > 0.55, n = 6/dose group).

**Figure 5 toxics-11-01027-f005:**
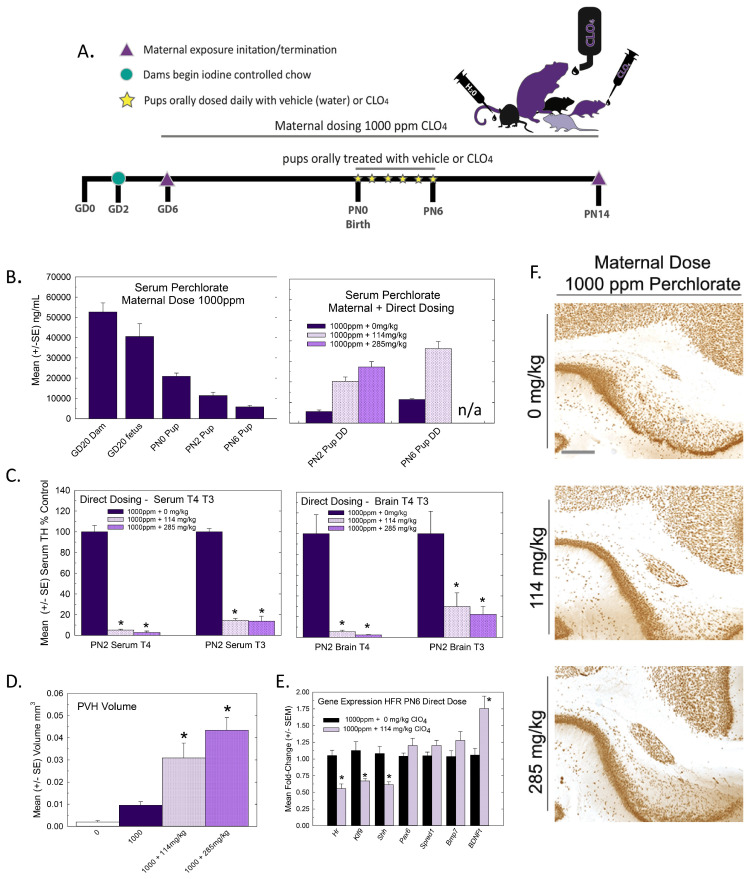
Experiment 2—Direct postnatal dosing with perchlorate. (**A**) Experimental design for Experiment 2. Pregnant rats were exposed to 1000 ppm of perchlorate in their drinking water beginning on GD6. At birth, pups from each litter were administered perchlorate orally by micropipette twice daily from PN0 to PN6. Serum and brain samples were collected from pups from each dose group on PN2 and PN6 for thyroid hormone analyses. An additional male pup was collected on PN6 for PVH gene expression. On PN14, the remaining pups were euthanized and their brains were collected for immunohistochemistry and processing for PVH. (**B**) Serum perchlorate (Left) from maternal exposure to 1000 ppm is high in the dam and the fetus but drops precipitously from birth over the first postnatal week. Direct dosing of offspring (right) from 1000 ppm-exposed dams increased serum perchlorate concentrations to approach those achieved in the fetus and newborn and maintained them at these levels until PN6. n/a, not available. (**C**) Serum T4 (F(2,35) = 31.8, *p* < 0.0001) and T3 (F(2.33) = 31.75, *p* < 0.0001) were reduced on PN2 at both concentrations of oral perchlorate supplementation (left). Similarly, brain T4 (F(2,6) = 27.64, *p* < 0.0009) and brain T3 (F(2,6) = 7.85, < 0.0211) were reduced on PN2 (Right). Sample size for serum, n = 8 for 0 and 114 mg/kg, n = 3 285 mg/kg; for brain, n = 3/dose group. * Dunnett’s *t*-test, *p* < 0.05. (**D**) Oral perchlorate supplementation dramatically increased the size of the PVH present at PN14. For reference, PVH volume in controls from Experiment 1 was added to the plot (clear bar labeled 0), supporting previous results of a small PVH following maternal exposure alone (1000 + 0 mg/kg), with dose-dependent increases in volume clearly induced with the direct dosing of pups. An ANOVA of the 0, 114 and 265 mg/kg dose groups revealed a significant effect of dose (F(2,18) = 10.57, *p* < 0.0009, n = 5–8/group). * Dunnett’s *t*-test, *p* < 0.05. (**E**) Offspring of dams exposed to 1000 ppm of perchlorate and orally supplemented with 0 or 114 mg/kg for the first 6 days of life displayed significant reductions in TH-responsive genes in the PVH-forming region (* *p* < 0.006, n = 7/dose group). (**F**) Photomicrographs of representative PVH images in brains of PN14 pups receiving 0, 114 or 285 mg/kg of perchlorate for the first 6 days of life against a background of 1000 ppm of maternal exposure. Scale bar 100 µm.

**Figure 6 toxics-11-01027-f006:**
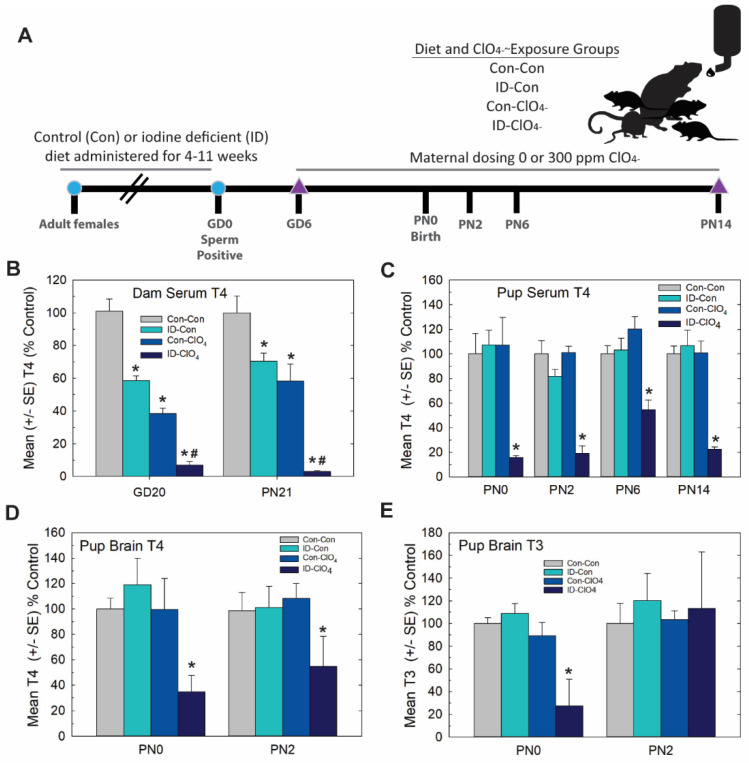
Experiment 3—Dietary iodine deficiency and perchlorate exposure. (**A**) Experimental design. Naïve female rats were administered an iodine-replete (Con) or iodine-deficient (ID) diet for 4–11 weeks before breeding. Sperm-positive females were then administered 0 or 300 ppm perchlorate (ClO_4_) in their drinking water, beginning on GD6 and continuing until pups were weaned on PN21. Pups were euthanized for serum and/or brain thyroid hormone evaluations on PN0, PN2, PN6 and PN14. Brains were collected on PN14 for immunohistochemistry. (**B**) Serum T4 was reduced in dams in late gestation (GD20) and at the weaning of pups (PN21) by an ID diet or 300 ppm of perchlorate in drinking water. Perchlorate administered to ID dams further reduced serum T4. An overall ANOVA revealed significant main effects of treatment (F(3, 96) = 81.4, *p* < 0.0001), with no effect of age or a treatment X age interaction (both *p*’s > 0.16). Step-down ANOVAs for treatment were significant at each age (all *p*’s < 0.0001), and Dunnett’s t-test supported a greater reduction in dam serum T4 with combined exposures. * *p* < 0.05 from Con-Con, # *p* < 0.05 from ID-Con or Con-ClO_4_, n = 10–15/treatment condition. (**C**) Pup serum T4 was only reduced in the ID-ClO_4_ condition at any age tested. An overall ANOVA revealed significant main effects of dose (F(3,160) = 52.57, *p* < 0.0001) and age (F(3,160) = 2.70, *p* < 0.0475) but no dose X age interaction (F(9,160) = 1.03, *p* > 0.42). Step-down ANOVAs at each age were significant (all *p*’s < 0.0003), with the ID- ClO_4_ group differing from Con-Con at all ages. (* *p* < 0.05 Dunnett’s *t*-test, N = 7–14/treatment condition). (**D**) Pup brain T4 was reduced; an overall ANOVA revealed significant main effect of dose (F(3,37) = 6.27, *p* < 0.0015), with no effect of age or dose X age interaction (both *p*’s > 0.72). A step-down ANOVA for brain T4 at PN0 was significant (F(3,19) = 4.80, *p* < 0.0119), but failed to reach significance at PN2 (F(3,18) = 1.98, *p* > 0.15). (**E**) An overall ANOVA for brain T3 did not yield statistically significant differences among groups. However, a step-down ANOVA for PN0 brain T3 was significant (F(3,19) = 6.59, *p* < 0.0031), confined to the ID- ClO_4_ treatment, while no differences in brain T3 were seen on PN2. Variability was high for both T3 and T4 the ID-ClO_4_ treatment and may have obscured the reliable detection of alterations in brain T3 on PN2. For D and E * *p* < 0.05 Dunnett’s *t*-test, N = 5–6/treatment condition at each age.

**Figure 7 toxics-11-01027-f007:**
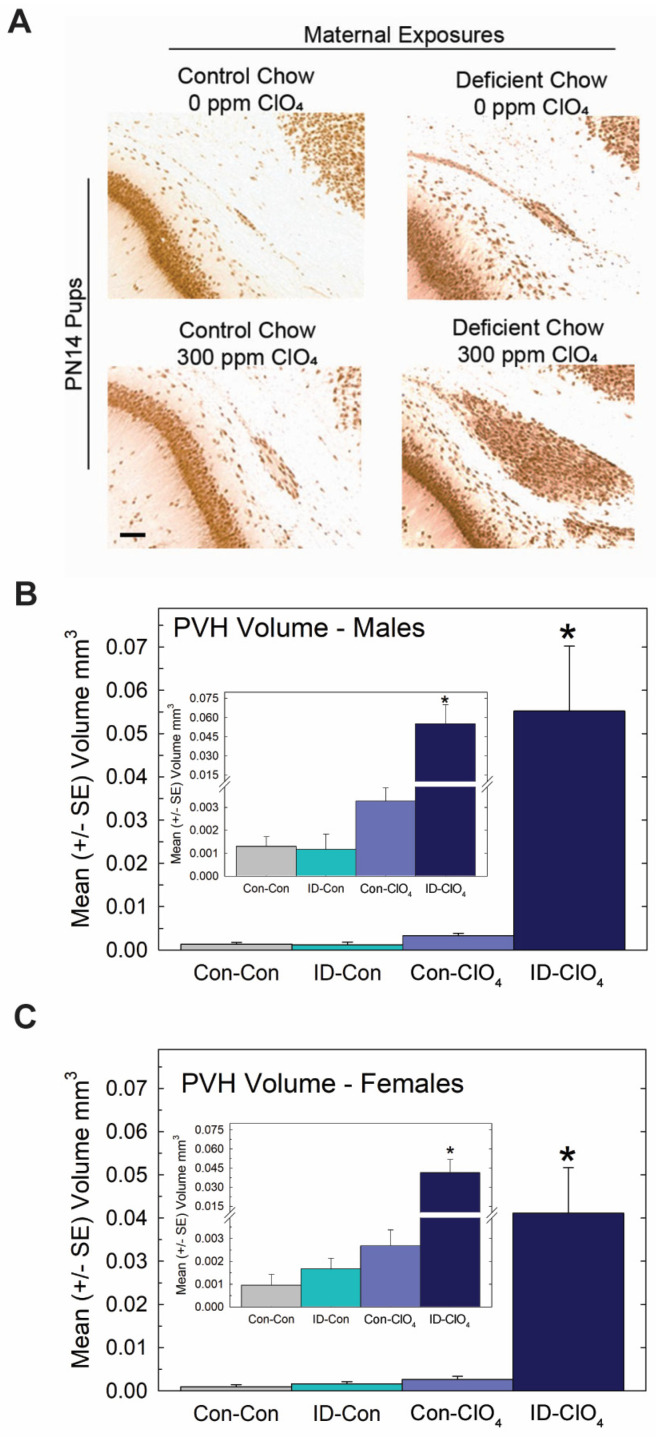
Periventricular heterotopia (PVH) were enlarged when perchlorate exposure occurred under conditions of dietary iodine deficiency. (**A**) Photomicrographs of a representative PVH in each treatment condition. PVH volumes in male (**B**) and female (**C**) offspring (n = 10–12 of each sex/treatment condition). An overall ANOVA returned a significant main effect of treatment (F(3,79) = 15.7, *p* < 0.0001) with no effect of sex or sex X treatment interaction (*p*’s > 0.89). * *p* < 0.05 Dunnett’s *t*-test after a significant one-way ANOVA for each sex. Inset shows a suggestion of increased PVH in the Con-ClO_4_ group, consistent with incidence data (see text) and the results of Experiment 1 at this dose level. However, in neither case was a statistically significant change in volume from background levels observed. Calibration bar 100 µm.

**Table 1 toxics-11-01027-t001:** Gene transcript names and IDs assessed in heterotopia-forming region of rat brain.

PVH Area Transcripts—PN6	Gene Name		ID
Direct TH Target	*Hr*	Hairless, HR Lysine Demethylase and Nuclear Receptor Corepressor	Rn00577605_m1
	*Shh*	Sonic hedge hog	Rn00568129_m1
	*Klf9 (Bteb1)*	Kruppel-like factor 9, Basic transcription element binding protein	Rn00589498_m1
TH-Responsive	*Bdnf_total_*	Brain-derived neurotropic factor	Rn02531967_s1
	*Bmp7*	Bone morphogenetic protein 7	Rn01528889_m1
PVH-Associated Transcripts	*Spred1*	Sprouty-related, EVH1 domain-containing protein 1	Rn01486390_m1
	*Pax6*	Paired box protein Pax-6	Rn00689608_m1
Reference Gene	*B2m*	Beta 2-microglobulin	Rn00560865_m1

## Data Availability

Data are contained within the article.

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
