# Peer review of "Structural Malformations in the Neonatal Rat Brain Accompany Developmental Exposure to Ammonium Perchlorate†"

_toxics, 2023, doi:10.3390/toxics11121027_

Round 1

Reviewer 1 Report

Comments and Suggestions for Authors

The present work is an important study that investigates the impact of a defined thyroid disruptor, perchlorate, on maternal/pup circulating and brain T4/T3, the development of a periventricular heterotopia (PVH), as well as related end-points of gene expression. The manuscript defines 3 distinct experimental conditions. The authors aim to expand their work on using the PVH as a readout of thyroid hormone disruption and test whether perchlorate, which disrupts thyroid hormone synthesis by inhibiting the sodium/iodide symporter (NIS), can induce a PVH as observed in inhibitors thyroid peroxidase (TPO).

There are several interesting and novel findings from this study. (1) Perchlorate alone at a moderate dose transiently in the perinatal period reduces serum and brain T3/T4, but induced PVH formation. (2) Extending perchlorate exposure to pups via direct pup dosing greatly enhanced effects on serum and brain T3/T4 and PVH formation. (3) Coupling a moderate perchlorate dose to low iodine diet additionally exacerbated the effect of perchlorate alone.

Overall, the study is well documented.  It provides important information for the regulatory domain and enhances our understanding of how thyroid disrupting chemicals may affect brain thyroid status and development. A few comments are provided that the authors might want to consider. 

1.     A bit more detail on how brain tissue was grossly dissected for gene expression studies would be appreciated. How was the tissue collection standardized? This seems important considering the relatively small size of the PVH compared to the overall brain size.

2.     It is noted that T4/T3 data reported in manuscripts under review is technically not previously reported. It is suggested that these statements be modified. (perhaps it is assumed that manuscript will be accepted/published prior to current manuscript)

3.     A similar point is noted with regard to perchlorate measurements.

4.     In the discussion (line 493), the manuscript states that it was demonstrated that the neonate was considerably more resilient relative to the fetal thyroid gland to the blocking effect of NIS by perchlorate... It is not clear how this was demonstrated. It is suggested that is idea, which is important, be clarified.

5. The observation that pup serum T4 significantly increased at P2 following an observed decrease at P0 is intriguing and likely an important observation. A similar pattern, although not significant, is observed in brain T4. How can this be explained? Perhaps, this observation should be included in the discussion.

Author Response

We thank the Reviewer for their support of our work. We have revised the manuscript according to their comments as follows.                                                                                                                         

  1. A bit more detail on how brain tissue was grossly dissected for gene expression studies would be appreciated. How was the tissue collection standardized? This seems important considering the relatively small size of the PVH compared to the overall brain size.
    1. We agree with the Reviewer and have expanded the text describing how this tissue was collected for the PVH-forming region used in gene expression experiments. This can be found on lines 176-184 in red font in the revised manuscript
  2. It is noted that T4/T3 data reported in manuscripts under review is technically not previously reported. It is suggested that these statements be modified. (perhaps it is assumed that manuscript will be accepted/published prior to current manuscript)
    1. This is exactly what has happened with a very efficient review of this manuscript by the journal Toxics, with a significant lag in the review of our first manuscript submitted to another journal. We have included methods for serum and brain hormone and perchlorate analysis into the current manuscript and eliminated reference to the 'under revision' citation. The additions can be found on lines 122-130 for serum perchlorate and lines 150-173 for serum and brain hormones, in red font, in the revised manuscript. We are happy to report that we have now just received word from the Editor and the manuscript in question is pending minor revisions so should also be out soon.
  3. A similar point is noted with regard to perchlorate measurements.
    1. Agreed - as above
  4. In the discussion (line 493), the manuscript states that it was demonstrated that the neonate was considerably more resilient relative to the fetal thyroid gland to the blocking effect of NIS by perchlorate... It is not clear how this was demonstrated. It is suggested that is idea, which is important, be clarified.
    1. On re-reading this section we fully agree with the Reviewer. We have now expanded our explanation in lines 505-518 in the discussion in red font in the revised manuscript. We based this conjecture largely on observation from our 'under review' manuscript (which will be published soon). In that paper we report a profound increase in Nis expression in the gland of pups from a 5-fold increase in the fetal gland to a 24-fold increase just 2 days later when the pups are born. Lower serum concentrations of perchlorate in the >PN2 neonate relative to the fetus and very large increases in Nis in the newborn are taken as evidence of greater resilience to the effects of perchlorate in the neonate in contrast to the late term fetus. 
  5. The observation that pup serum T4 significantly increased at P2 following an observed decrease at P0 is intriguing and likely an important observation. A similar pattern, although not significant, is observed in brain T4. How can this be explained? Perhaps, this observation should be included in the discussion.
    1. We thank the Reviewer for this recommendation. We have highlighted this observation in the Results in lines 270-272 and in the Discussion in lines 515-518. We offer in this text that an interaction of augmented Nis expression in the gland in the face of falling perchlorate levels in the serum of the pups may be responsible for a transient overshoot in hormone as the thyroid system of the very young neonate attempts to achieve homeostatic balance. 

Reviewer 2 Report

Comments and Suggestions for Authors

The paper by Dr Gilbert and collaborators describes the neurodevelopmental effects of perchlorate, an environmental chemical present in drinking water and acting as an inhibitor of iodine uptake into the thyroid gland leading to decreased levels of thyroxine (T4) synthesis. In particular, by using a rat pregnant model and administering different doses of perchlorate in the drinking water of dams from GD6 to postnatal day 21, the author demonstrated a dose-dependent decrease of T4 in the dam serum during the gestational phase and beyond, but only at P0 in the pup serum using the highest doses, suggesting a recovering capacity of the pups from PN2 onwards. Similarly, dose-dependent reduction of thyroids hormone T3 and T4 was observed in the brain of pups at GD20 and at birth (PN0), but not later. Accordingly, no (or small) periventricular heterotopia (PVH), a cluster of ectopic neurons generating in the posterior forebrain of animals born from hypothyroid dams (and associated gene expression), was observed.

On the contrary, when pups were exposed to direct dosing of perchlorate along with the highest maternal supplementation dose, this combined treatment induced a drastic increase in the size of PVH at PN14 and a significant reduction of T4 and T3 in both serum and brain of PN2 pups.

In addition, when dams were treated with a lower amount of perchlorate from GD6 to PN14 but simultaneously subjected to an iodine deficiency diet for 4-11 weeks, a strong reduction of T4 dam serum during the whole treatment period (GD20-PN21) and a concurrent reduction of T4 pup serum from PN0 to PN14 were observed. T3/T4 pup brain reduction was also revealed at PN0 and only T4 at PN2. These alterations were accompanied by a large increase in PVH volume, with no differences between sexes.

Overall, these series of studies demonstrated a perchlorate toxic effect on brain development, particularly when maternal exposure to this chemical is combined with a direct pup administration or with a maternal iodine deficiency diet, highlighting that chemical and non-chemical environmental stressors have to be considered in the estimation of risk for chemical hazard and that chemicals that not only perturb the maternal hormone profile but that can have access to the fetus increase the risk for brain development. 

Although extrapolating these data to evaluate the risks for human brain development is not easy, the paper is very interesting, the experimental plan is well organized and the results are consistent with previously published data from the same group and others and are reliable. The result discussion is clear.  

However, to further improve paper quality I would suggest some additional experiments and revisions, as follows:

-Fig. 4 A shows negative results in the expression of genes related to PVH, according to the small size of this anatomical formation in response to maternal treatment while Fig.4B reports the effects on gene expression of the oral dosing of the pups, so it should be shown after Fig. 5 and not anticipated in Fig.4;

-Relatively to gene expression analysis, it would be interesting to evaluate gene expression changes in response to the combined treatment of maternal iodine restrictive diet/perchlorate administration which causes the large PVH formation shown in Fig. 7;

-Negative gene expression results (Fig.4A) could be transferred to supplementary data;

-Since the paper mentioned as “Gilbert under review” has not been published yet, the methodology used in that paper has to be briefly reported in the MM section where the paper is quoted (line 132, pag.3);  

-The section from lines 80 to 87 of the Introduction should be transferred to the Discussion since it is a summary and a discussion of the results obtained.

- In the text references must be indicated in numbers

Author Response

We thank the Reviewer for their comments and support of our findings. We have revised the manuscript according to their recommendations as follows.

  1. Fig. 4 A shows negative results in the expression of genes related to PVH, according to the small size of this anatomical formation in response to maternal treatment while Fig.4B reports the effects on gene expression of the oral dosing of the pups, so it should be shown after Fig. 5 and not anticipated in Fig.4
    1. We agree with the Reviewer and have presented the negative findings on gene expression in the PVH-forming region from Experiment 1 as a Figure unto itself. We chose to keep these data in the main part o f the manuscript rather than report them as supplementary material as they make the case stronger for the relative absence vs robust effects when the system is further perturbed int he postnatal period. We have incorporated the positive gene expression findings of the PVH-forming region in the Direct Dosing into Figure 5 where all the results of Experiment are summarized. 

2.  Relatively to gene expression analysis, it would be interesting to evaluate gene expression changes in response to the combined treatment of maternal iodine restrictive diet/perchlorate administration which causes the large PVH formation shown in Fig. 7

   We agree that gene expression from Experiment 3 would have been interesting, but unfortunately this tissue was not collected in the ID/Perchlorate study

3. Since the paper mentioned as “Gilbert under review” has not been published yet, the methodology used in that paper has to be briefly reported in the MM section where the paper is quoted (line 132, pag.3);  

      We agree with this Reviewer and Reviewer 1 who made the same recommendation. We have removed reference to the manuscript that is yet to be published and incorporated methodology for serum perchlorate and thyroid hormone analyses into the methods of this paper. These can be found on lines 122-130 and Lines 150-173 in the revised Methods in red font.

4. The section from lines 80 to 87 of the Introduction should be transferred to the Discussion since it is a summary and a discussion of the results obtained

   We understand the Reviewers point but chose to maintain the summary of the 3 studies that appears in the introduction as it orients the reader to the three studies that are to be reported. 

5.  In the text references must be indicated in numbers

    We thank the reviewer for this notification! We entirely missed it in preparation of this submission. We have reformatted the  reference section according to journal specifications.